# Placental malaria caused by *Plasmodium vivax* or *P. falciparum* in Colombia: Histopathology and mediators in placental processes

**Jaime Carmona-Fonseca[1], Jaiberth Antonio Cardona-Arias**[1,2]*

**1** "Grupo de investigación César Uribe Piedrahíta", Faculty of Medicine, University of Antioquia, Medellín, Colombia, **2** School of Microbiology, University of Antioquia, Medellín, Colombia

* jaiberth.cardona@udea.edu.co

**Data Availability Statement:** All relevant data are within the paper and its Supporting information files.

## Abstract

Knowledge about the relation of histopathological characteristics and mediators of physiological processes in the placenta malaria (PM) is poor, and that PM caused by *Plasmodium vivax* is almost null. The objective was to compare histopathological characteristics, cytokines and mediators of physiological processes in PM depending on the parasitic species, through a cross-sectional study in three groups: negative-PM, *vivax*-PM, *falciparum*-PM from Northwestern Colombia. The diagnosis of PM was made with thick blood smear, qPCR, and histopathology. Immuno-histochemical was made with EnVision system (Dako) and Zeiss Axio Imager M2 with light microscope. Cells in apoptosis were studied with the TUNEL technique. To measure the expression level of cytokines and mediators qRT-PCR was used. We included 179 placentas without PM and 87 with PM (53% *P. vivax* and 47% *P. falciparum*). At delivery, anemia was 25% in negative-PM, 60% in *vivax*-PM, and 44% in *falciparum*-PM group. The neonatal weight had an intense difference between groups with 3292±394g in negative-PM, 2,841±239 in *vivax*-PM, and 2,957±352 in *falciparum*-PM. The histopathological characteristics and CD+ cells in placenta with statistical differences (Dunn's test) between negative-PM vs *vivax*-PM (*P. falciparum* was similar to *P. vivax*) were infarction, fibrinoid deposits, calcification, cells in apoptosis, immune infiltrates in decidua and intervillous space, CD4+, CD8+, CD14+, CD56+, CD68+. The expression levels of mediators in the placenta with statistical differences (Dunn's test) between negative-PM vs *vivax*-PM (*P. falciparum* was similar to *P. vivax*) were Fas, FasL, HIF1α, Cox1, Cox2, VEGF, IL4, IL10, IFNγ, TNF, TGFβ, FOXP3, and CTLA4. PM with *P. falciparum* and *P. vivax*, damages this organ and causes significant alteration of various physiological processes, which cause maternal anemia and a reduction in neonatal weight in degrees that are statistically and clinically significant. It is necessary that the search for plasmodial infection in pregnant and placenta goes from passive to active surveillance with adequate diagnostic capacity.

**Funding:** 1) Colciencias project 111 577 757 447, contract 755-2017. Jaime Carmona-Fonseca. 2) Universidad de Antioquia. Jaiberth Antonio Cardona Arias and Jaime Carmona-Fonseca received a salary from this funder. The funders had no role in study design, data collection and analysis, decision to publish, or preparation of the manuscript.

**Competing interests:** The authors have declared that no competing interests exist.

## Introduction

Malaria associated with pregnancy (MAP) or malaria in pregnancy (MiP) can manifest itself in one, two, or all three of the following forms: gestational malaria (GM) in the mother, placental malaria (PM) in the placenta, or congenital malaria (CM) in the newborn. The knowledge on MiP is much more advanced for *Plasmodium falciparum*, while it is poorly developed for *P. vivax*. Plasmodial infections of MiP can be microscopic (detected with microscopy, that is, with thick blood smear [TBS]) or submicroscopic (not detected with microscopy but with another test, usually a technique of amplification of nucleic acids of *Plasmodium*, such as the polymerase chain reaction PCR). Both microscopic and submicroscopic plasmodial infections (SPI) compromise the health of the pregnant woman, the placenta, and the gestational product [1–5].

MiP caused by *P. falciparum* (*falciparum*-PM) is a fully established entity with great importance as a health problem [6–8]. MiP caused by *P. vivax* (*vivax*-PM) has been known for years, but its systematic study has hardly happened in the last 20 years; however, it is still necessary to know the bases of pathogenesis and the histopathological and immunopathological aspects.

In PM's immune response, the immune cells play a crucial role, but little known in a specific way. The class, quantity, and location of these cells need to be identified to clearly understand the entity's pathogenesis. Something similar happens with the cytokines and mediators of physiological processes involved (apoptosis, hypoxia, inflammation, angiogenesis, etc.). The endemicity of malaria in America and Southeast Asia does not reach the typical levels of many countries in Sub-Saharan Africa, and another epidemiological issue of great interest is *P. vivax* and *P. falciparum*'s coexistence, with *P. vivax* predominating. We know that *vivax*-MiP with high frequency is submicroscopic and affects the pregnant woman, the placenta and the gestational product [1–5, 9, 10].

Based on this context, the study objectives were to seek a comprehensive approach to PM depending on the parasitic species and taking into account the placental tissue characteristics and the magnitude of expression of the genes that control some cytokines and mediators of physiological processes in the placenta.

## Methods

### Study location

Women were recruited between 2004–2019 at the hospital obstetric facilities in the municipalities of Puerto Libertador and Tierralta in the Department of Cordoba, and Necoclí, Turbo and Carepa in Urabá Antioqueño, in the Department of Antioquia. The municipalities are part of the region formed by "Urabá Antioqueño-the upper basins of the Sinú and San Jorge rivers in the south of the Department of Córdoba-Bajo Cauca Antioqueño" in northwestern Colombia (from now on "The Region") [11].

"The Region" is homogeneous in terms of ecoepidemiology and malaria transmission. Transmission intensity is medium to high, with no marked fluctuations in the number of malaria cases during the year (stable transmission). This region has an estimated area of 43,506 km2, a malaria risk population of 1.1 million in 2018–2019, with a mean annual parasite index of 35.8 cases/1,000 inhabitants. Most municipalities in "The Region" have a high endemicity of malaria all year [11]. In "The Region" and in Colombia in general, there is no problem of *Plasmodium* resistance to the antimalarial treatments adopted or of the anopheline vectors to the insecticides used to control transmission [10, 12].

## Sample design and size

A analytical cross-sectional study was used to compare three groups: negative-PM, *vivax*-PM and *falciparum*-PM. To control for variations in the behavior of the histological characteristics and the process mediators in the placenta, it was defined that only term pregnancies and placentas (37.0–42.0 weeks) are considered. Placentas with plasmodial infection were obtained between 2004 to 2019. Mixed infection was not included (the frequency of mixed infection in this area of Colombia is 1.44% on average, in the general population and according to the thick blood smear) [13, 14].

Due to the absence of previous studies that report the difference in means in the variables analyzed in this study (between the infected and non-infected group), and taking into account the high number of variables compared, the sample size of the comparisons made in this study fits the following parameters: difference of standardized means (disappearing the units of measure of each variable compared) of 0.5; power of 0.95; 95% confidence and non-infected-infected ratio 2. For financial reasons, the immunohistochemical, cytokine, and mediators of physiological processes studies were performed in a subsample of 75 units (25 with *P. vivax*, 25 with *P. falciparum*, 25 without infection); this size was arbitrarily defined and was randomly taken from the respective group in the biological sample bank.

## Inclusion and exclusion criteria

The criteria applied were based on:

*Inclusion*: a) Being a permanent resident in a malaria-endemic region at least for the year prior; b) Having delivered in one of the local hospitals; c) Voluntary acceptance to participate in the study.

*Exclusion*: a) Having other concomitant infections with malaria, different from intestinal parasites; b) Having pre-eclampsia/eclampsia, hypertensive disorder of pregnancy, or uncontrolled diseases like diabetes, asthma, high blood pressure, etc, c) Appearance of a complication of malaria or other diseases; d) Withdrawal of consent.

## Sources and methods of selection of participants

Women were recruited at the hospital obstetric facilities in the municipalities already indicated. These hospitals belong to the public network and mainly serve low-income residents. The hospitals offer prenatal consultation and delivery care to the inhabitants of urban and rural areas. The costs of the services are paid by insurance companies or by the Colombian state. Women were included in the order in which they presented to the hospital and there was no special incentive to attend the hospital or to participate in the investigations.

## Data collection

After inclusion, a form was completed to obtain information from each woman through a questionnaire and consulting her medical history, prenatal registry, and delivery registry. Demographic, clinical, and epidemiological information was obtained on the mother and the newborn.

**Specimen collection for diagnosis.** All the procedures of the histological, immunohistochemical and gene expression studies were carried out by members of the research group that has carried out the different research projects. The authors of this report proposed and executed the analytical approach to the data for actual document.

**Maternal peripheral blood and placental blood.** The elaboration, coloring, and reading of the TBS was done according to the procedure recommended by the WHO. The blood sample was taken by venipuncture (≈10 mL) in a tube with EDTA (ethylenediaminetetraacetic acid) and a dry red-top tube. Placental blood was obtained from the lake formed after a wash with saline (0.9%) and removal a fragments taken close to the insertion point of the umbilical cord and in the middle area (equidistant between the cord and the placental edge). They were used to 1. make thick and thin blood smears for diagnosis of malaria; 2. prepare two Whatman filter paper circles # 3 with approximately 100 uL of blood in each, which were used for the molecular diagnosis of malaria; 3. determine the levels of ferritin and albumin.

**Placental blood for quantitative polymerase chain reaction.** A hole punch circle (≈6 mm) from each filter paper was used for DNA extraction, corresponding to approximately 25 μL blood. Widely known protocols were followed [5, 15–17].

**Placental tissue.** A hole punch circle (≈6 mm) from each filter paper was used for DNA extraction, corresponding to approximately 25 μL blood [5]. DNA was extracted with the saponin-Chelex method. The extracted DNA was resuspended in 50 mL water. Real-time quantitative PCR (qPCR) was performed as described in a subsequent section (Shokoples et al., 2009). Samples were first tested for *Plasmodium* using genus-specific primers and a hydrolysis probe (Plasprobe). PCR was run on the ABI 7500 FAST platform. Samples with a Cycle Threshold (Ct) <45 were tested with duplex species-specific reactions for *P. falciparum* and *P. vivax*. In the tests, amplification of the 18S rRNA genes for DNA for quantification was used. DNA copy number was quantified from the genus-specific reaction against a standard curve using a plasmid containing a fragment of the 18S gene from *P. falciparum* [5].

**Histopathology.** All placental biopsy specimens were examined by trained personnel without prior knowledge of the maternal characteristics, pregnancy outcome, or malaria episodes in pregnancy. Paraffin-embedded placental specimens were cut to produce at least three sections at a thickness of 5 mm, stained with hematoxylin and eosin, and examined by microscopy under 40X (magnification = 400) and 100X (magnification = 1,000; high- power field) [5, 10].

Histologic events were evaluated in two ways: qualitatively (present vs. absent) and quantitatively (quantity). For each variable, the absence of the event corresponded to a value of zero in all 40 microscopic fields examined; the presence of the event in at least one field was considered as the presence of the event. The quantitative evaluation was done in two ways: a) for events abruption, atherosis, decidual necrosis, villous edema, villous infarction, villous hemorrhage, thrombi in IVS, and calcifications in IVS, the amount of event was added in each of the 40 fields, and the total sum was considered as the amount of the event; b) for events fibrinoid deposits, syncytial nodes, placental villi, fetal capillaries, capillaries per villus, and immune cells, the amount of event was added in each of the 40 fields and the total sum was divided by 40; the average obtained was considered as the amount of the event [5]. (S1 Text).

**Immuno-histochemical study.** In formalin-fixed, paraffin embedded tissues, the EnVision system (Dako) with anti-Human CD4 (Clone 4B12, Dako), anti-Human CD8 (Clone C8/144B, Dako), anti-Human CD14 (Clone TÜK4, Dako), anti-Human CD56 (Clone 123C3), and anti-Human CD68 (Clone PG-M1, Dako), was used. Briefly, the paraffin sections were deparaffinized and rehydrated in xylene and graded alcohols. After blocking with peroxidase in ChemMate peroxidase-blocking solution (Dako), the slides were incubated with the primary antibodies. After application of the peroxidase-labeled polymer, the slides were incubated with the diaminobenzidine substrate chromogen solution, counterstained with hematoxylin, washed again, dehydrated, and mounted. The immunohistochemical slides were observed using a Zeiss Axio Imager M2 light microscope equipped with a Zeiss Axio Cam HRc Camera to capture images of the placenta. Ten photos were collected per slide, with 40X objective lens. Subsequently, each photograph was analyzed and the number of cells counted

for each photo. The number of positive cells was calculated and analyzed using the Image J software (Image J 1.46r Wayne Rasband National Institutes of Health, USA, https://imagej.nih.gov/ij/index.html).

**Quantification of cells in pre-apoptosis and apoptosis.** Apoptosis was studied with the TUNEL technique (Terminal deoxynucleotidyl transferase dUTP nick end labeling) [18]. 3 μm cuts were made from paraffin blocks of placental tissue, and Superfrost Plus™ Microscope Slides (Thermo Scientific) were prepared. The tissue was deparaffinized with xylol and ethanol at different concentrations and subsequently paraformaldehyde to fix it to the sheet.

The DeadEnd™ Colorimetric TUNEL System reagent set (Promega) was used for the colorimetric detection of apoptotic cells in the tissue fixed on the slide. The slides were viewed with a Zeiss Axio Imager M2 optical microscope equipped with a Zeiss AxioCam HRc camera, which was used to capture images of the placenta. The number of positive cells, which were a brown color, was calculated with the Image J program (J Image 1.46r Wayne Rasband National Institute of Health, USA, https://imagej.nih.gov/ij/index.html).

The percentage ratio of apoptotic cells was obtained by taking into account the number of positive nuclei (colored brown) and the number of total cells observed in 40 fields with a 40X objective; The number of TUNEL positive cells was divided by the total cells counted and multiplied by 100 to find the percentage.

**Measure the expression level of cytokines and mediators.** A reverse-transcription real-time PCR assay with relative quantification (qRT-PCR) was used to evaluate the expression of immune mediators of inflammation (TNF, IFN-γ, IL-8 and CD54), anti-inflammation (IL-10, TGF-β and IL-13), regulation (FOXP3, CTLA4, TNFRII, CD163 and PD-L1) and co-stimulation (CD86 and CD40) of the immune response in placental tissue. Each sample per molecule was evaluated in triplicate. Total RNA was extracted using QIAamp RNA Blood Mini® (QIAGEN) from the placental tissue stored at 4 ˚C in RNALater® (Qiagen), taking into account the manufacturer's instructions. The EXPRESS OneStep Superscript® qRT-PCR kit (Invitrogen) was used for reverse transcription and amplification of each molecule, using a StepOnePlus real-time PCR system (Applied Biosystem). The OneStep Software V2.3 was used for data analysis. Relative quantification was calculated using the (2— ΔΔCt) method [19] and normalized with β-actin as the reference gene. A pool of RNA of placental tissue from healthy pregnant women was used as a calibrator. All experiments included a no-template control.

## Statistical analysis

The variables do not have normal distribution according to the Kolmogorov-Smirnov test. The Kruskal-Wallis (K-W) was used to compare medians of histopathological characteristics and cytokines and mediators of physiological processes in the three study groups. Dunn's test for multiple comparisons was used to identify the pairs of groups that differ statistically when the K-W test indicated that there was a difference in the three groups compared.

The association between two qualitative variables was assessed with the chi-squared test ($X^2$); in case of finding an association between the qualitative variables and the study group, in the comparisons (multiple) between each pair (negative-PM Vs *vivax*-PM; negative-PM Vs *falciparum*-PM, and *vivax*-PM vs *falciparum*-PM), the p-value was adjusted with the Bonferroni test. The correlation between two variables was measured with Spearman's Rho (Spearman's Rank Correlation Coefficient). The analyzes were done with the SPSS 25.0.

## Ethical considerations

The project received the endorsement of the Bioethics Committee of the SIU (University Research Headquarters), Universidad de Antioquia, approval certificate 07-32-12. All

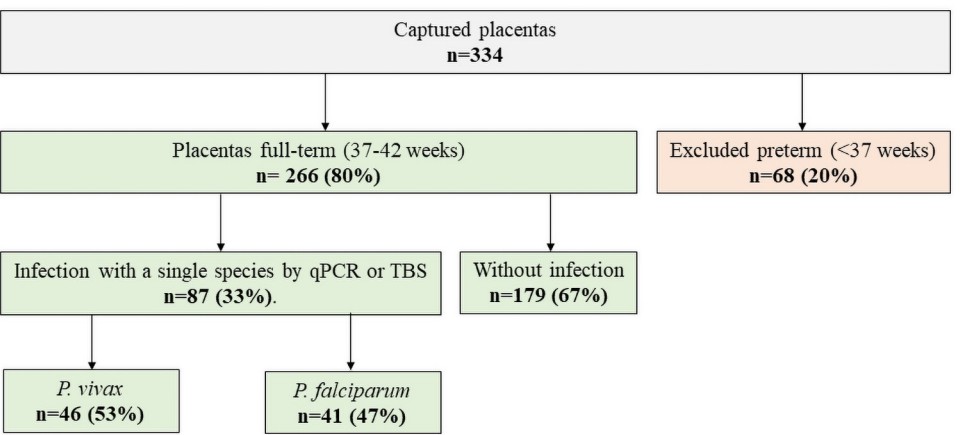

**Fig 1. Flowchart on the selection process of the 266 placentas finally analyzed.**

participants signed the consent (of legal age) or assent (under 18 years of age) informed, obtained in writing. Colombian law allows that adolescents (over 10 years old) sign the informed assent without consent from parents or guardians. All the data were anonymized by someone outside the authors of this manuscript.

# Result

We obtained and included in the analysis 266 units (mother-placenta full-term): 179 without PM, 46 with *P. vivax* and 41 with *P. falciparum* (Fig 1).

## General characteristics of the analysis units

The 266 women are young (negative-PM: 23.1±6.2 years; positive-PM: 24.8±6.3 years) (Table 1). They have had between 2.7 and 3.3 deliveries; The number of pregnancies is as follows: in the first pregnancy 34%, in the second pregnancy 20%, in the third 16%, in the fourth

**Table 1. General characteristics of the analysis units according to the study group.**

| Variable | Negative | *P. vivax* | *P. falciparum* | p K-W |
|---|---|---|---|---|
| | n = 179 | n = 46 | n = 41 | |
| | Median (Min-Max) | | | |
| Age (years) | 22.0 (13–42) | 23.5 (14–39) | 25.0 (15–38) | >0,05 |
| Parity | 2.0 (1–10) | 2.0 (1–7) | 3.0 (1–8) | >0,05 |
| Month of episode [a] | 5.0 (1–8.8) | 3.5 (2–9) | 5.0 (2–8) | >0,05 |
| Gestation weeks (by DLM) | 38.8 (38–42) | 38.8 (38–41) | 39.0 (38–41) | >0,05 |
| Maternal Hb at delivery (g/dL) | 11.3 (7.8–18.3) | 11.2 (8.1–12.4) | 11.0 (9.0–12.6) | **0,004**[b] |
| Neonatal weight (kg) | 3.1 (2.2–4.6) | 3.0 (2.5–3.3) | 2.9 (2.5–3.8) | **≤0,001**[b] |
| | Percentage | | | p X$^2$ |
| Maternal anemia at delivery | 25 | 60 | 44 | **0,006** [c] |
| Low birth weight | 1,1 | 0,0 | 3,7 | **0,010** |

[a] Month of the current pregnancy when the malaria episode occurred.

[b] Dunn's test indicates a significant difference between negative-PM and *P.vivax*.

[c] Bonferroni's test indicates a significant difference statistical between all pairs.

12% and in the fifth to tenth 17%. They reported having had malaria in the current pregnancy 23% of the times, and the malaria episode occurred in the current pregnancy at 4.7 to 5.5 months of gestation. Delivery occurred at 39.2–39.3 weeks' gestation (according to date of last menstruation, DLM).

Maternal hemoglobin (Hb) at delivery was 10.6±1.1 to 12.6 g/dL, with difference between *vivax*-PM and negative-PM, but not between *vivax*-PM and *falciparum*-PM; this means that *vivax*-PM was associated at Hb level as much as *falciparum*-PM did. Also, there was no significant difference between negative-PM and *falciparum*-PM (Table 1).

At the time of delivery, anemia (<11.0 g/dL) was present in 25% of women in the negative-PM group, in 60% in the *vivax*-PM group, and in 44% in the *falciparum*-PM group, with a significant difference between the three groups (p($X^2$ Pearson) = 0.006) and between the negative-PM and *vivax*-PM but not between the two groups with infection (p($X^2$ Fisher) = 0.222).

The neonatal weight had difference between the three groups and between the two groups with infection. On average there was a 335g difference between the control group and the one infected with *P*. *falciparum* and 451g between the control and *P. vivax*. These differences are clinically significant, whether or not there is statistical significance, which also exists between negative-PM and *vivax*-PM. Low birth weight (LBW) occurred in two cases (1.1%) of the negative-PM group and one case (3.7%) of the *falciparum*-PM. Of these variables outlined, only maternal Hb at delivery and neonatal weight present a statistically significant difference between the three groups (Table 1).

## Histopathological events in PM according to Plasmodium species

The histopathological events showing a significant difference between the three groups are infarction, fibrinoid deposits, syncytial nodes, capillaries, capillaries per villus, calcifications, immune cells in the decidua, villi, and IVS; and all CD+ cells. All variables with a significant difference have higher amounts when there is infection. Capillaries and capillaries per villus have higher values in the presence of PM. Dunn's test indicates significant differences between negative-PM and *vivax*-PM for the events infarction, fibrinoid deposits, calcifications, CID, CIEV, CD4, CD8, CD14, CD56, and CD68, but there is only a significant difference between *vivax*-PM and *falciparum*-PM in the variable CD68 (Table 2).

In general, all these events are normal in the process of pregnancy and childbirth, except the presence of parasites or hemozoin; therefore, the critical issue is to compare the amount of the event with respect to the group without infection (negative-PM). Atherosis and necrosis have a higher amount in the absence of infection, but the difference is not significant. It should be noted that the number of these events is "low" in all groups, and the number of abruption and thrombi is also scarce. Edema, infarction, calcifications, and hemorrhages present a "moderate" amount in the three groups, and the first three events show a significant difference between the three groups. There is no difference between *vivax*-PM and *falciparum*-PM in edema, but there is a difference for the events infarction and calcifications. Fibrinoid deposits, syncytial nodes, villi, capillaries, capillaries per villus, immune cellular infiltrates, and CD+ specific immune cells have amounts ranging from relatively "low" to "high," and all these eight variables show a significant difference between the three groups. However, between *vivax*-PM and *falciparum*-PM, only in the case of CD68, there is a difference (Table 2).

The number of immune cells (mononuclear and polymorphonuclear) in each placental area (decidua, villi, intervillous space IVS) shows a significant difference between the three groups (Table 2). Within each placental area, depending on the presence of the plasmodial infection, the number of infiltrated immune cells behaves as follows:

**Table 2. Histopathological characteristics in placenta according to PM group.**

| Variable | Negative | *P. vivax* | *P. falciparum* | p K-W |
|---|---|---|---|---|
| | | Median (Min-Max | | |
| Atherosis | 0(0–5) | 0(0–4) | 0(0–4) | 0.445 |
| Abruptio | 0(0–12) | 0(0–10) | 0(0–12) | 0.154 |
| Necrosis | 0(0–13) | 0(0–1) | 0(0–0) | 0.414 |
| Edema | 1(0–26) | 4(0–20) | 6(0–21) | 0.095 |
| Villi | 315 (146–752) | 346 (193–397) | 335 (193–432) | 0.568 |
| Hemorrhage | 11 (0–32) | 9 (0–27) | 10 (0–27) | 0.778 |
| Thrombi | 0(0–10) | 0(0–9) | 0(0–9) | 0.394 |
| Infarction | 5(0–37) | 13(0–28) | 7(0–27) | **0.018** [a] |
| Fibrinoid deposits | 51(0–130) | 63(24–154) | 75(0–147) | **≤0,001** [a] |
| Syncitial nodes | 91(0–346) | 120(63–193) | 121(0–289) | **0.004** |
| Capillaries (Thousands) | 1.9(1.1–6.0) | 2.1(1.1–4.2) | 2.5(1.2–6.0) | **0.044** [a] |
| Capillaries per villus | 6(3–14) | 6(4–12) | 7(4–14) | **0.003** |
| Calcification | 0(0–16) | 3(0–32) | 4(0–37) | **≤0,001** [a] |
| CID | 6 (2–97) | 43 (29–126) | 57 (3–137) | **≤0,001** [a] |
| CIV | 25(0–143) | 34 (4–85) | 38 (0–119) | **≤0,001** |
| CIEV | 88 (10–596) | 118 (63–255) | 131 (12–319) | **≤0,001** [a] |
| CD4 Helper T lymphocytes | 14 (10–13) | 15 (12–17) | 15 (11–21) | **≤0,001** [a] |
| CD8 Cytolytic T lymphocytes | 11 (8–11) | 12 (11–15) | 12 (11–14) | **≤0,001** [a] |
| CD14 Monocytes | 51 (26–37) | 52 (49–61) | 53 (50–59) | **≤0,001** [a] |
| CD56 NK cells | 94 (70–87) | 98 (87–116) | 98 (91–114) | **≤0,001** [a] |
| CD68 Macrophages | 86 (40–48) | 87(81–91) | 93 (86–98) | **≤0,001** [a] |

[a] Dunn´s test with p<0.05 to compare PM negative vs PM by *P. vivax*.

1. When there is no PM, the number of immune cells/40 fields, forming infiltrates in each evaluated placental area, is higher in the IVS (109 cells), followed by the villi (30 cells) and, finally, the decidua (6 cells). The ratio between the averages of each zone indicates that in the villi, immune cells are five times those of the decidua, and in the IVS, they are 18 times those of the decidua.

2. When there is *vivax*-PM, the number of cells in the immune infiltrates is more significant in the IVS, but it is followed by the decidua and not by the villi as it happened in the absence of infection; the least amount of infiltrated cells are in the villi. The ratio between the averages of each zone indicates that, in the decidua, the immune cells are 1.8 times those of the villi, and in the IVS, they are 3.8 times those of the villi.

3. When there is *falciparum*-PM, the number of cells in immune infiltrates is higher in the IVS, then the decidua, and, finally, the villi, meaning it follows the same order as in *vivax*-PM. The ratio between each zone's averages is very similar to that of *vivax*-PM: 1.8 and 3.8 cells/40 fields, in decidua and IVS compared to villi.

The specific immune cells (CD+) behaved as follows: each of the five cell types shows a significant difference between the three groups, but they do not differ according to the species, except for CD68. The CD+ cells with the greatest abundance are CD68 (macrophages), followed by CD56 (NK), then CD14 (monocytes), CD4 (helper T lymphocytes), and finally CD8 (cytotoxic T lymphocytes). The ratio between the quantity of these cells, evaluated without discriminating by the state of plasmodial infection and based on CD8s, is 1.0 (for CD8), 1.9 for

CD4, 6.9 for CD14, 8.5 for CD 56, and 13.6 for CD68. CD8 cells are the rarest and other CD + cells are 2 to 13 times the number of CD8 cells.

According to histopathology, the inflammatory process, evaluated by the amount of immune infiltrates, is more intense when *falciparum*-PM is present compared to *vivax*-PM. CD+ specific immune cells do not show significant differences by species, and the amount of each of them is very similar in each of the species. The increase in CD+ cells per 40 fields between negative-PM and positive-PM is very intense for CD14 (it goes from 32 to 53 cells; an increase of 65%), CD56 (it goes from 77 to 102 cells; an increase of 32%), and CD68 (it goes from 44 to 90 cells; an increase of 104%).

The hemozoin (Hz) was found, through histopathology, in 20 placentas: 40% associated with *P. vivax*, with an average amount of 11 deposits in the 40 fields (always p = $\leq$0.001); 60% associated with *P. falciparum*, with an average amount of 15 deposits in the 40 fields. The infected erythrocytes (IEs) were found in three placentas: one with *P. vivax* and two with *P. falciparum*; IEs were associated with Hz in all three placentas; these IEs were found in IVS, in villi, or both. In the PM-no and *vivax*-PM groups, hemozoin did not present SC with any of the eight variables that express immune cells (infiltrates in decidua, villi and IVS, five classes of CD+ cells); in the *falciparum*-PM group, there was only one SC, positive, between Hz and the infiltrates in the villi (p = 0.03).

## Expression of cytokines and mediators of processes in PM according to Plasmodium species

The apoptosis process is expressed in healthy placentas, but at a lower level than in infected ones, with a strongly significant difference. Inflammation is present in placentas without infection, but it is undoubtedly significantly higher when there is infection; This is clearly expressed by the IFN$\gamma$, TNF, Cox1, Cox2, and VEGF mediators, although the levels of IL2, C5a, and VEGFR do not differ from those found between infection and non-infection (Table 3).

The angiogenesis process, judged by what happens with VEGF and VEGFR, was slightly increased in the infectious state's presence. Hypoxia was also significantly increased when infection occurs, as indicated by HIF1$\alpha$. Regulatory mediators (TGF$\beta$, FOXP3, and CTLA4) appear significantly increased in the presence of plasmodial infection. Six cytokines (IL2, IL4, IL10, IFN$\gamma$, TNF, TGF$\beta$) and two mediators (FOXP3 or scurfin [master gene controlling the development and function of regulatory cells], CTLA4 [antigen-4 associated with cytotoxic T-lymphocyte; it is a molecule that inhibits activation of T lymphocytes]) were evaluated in the placenta and, except for IL2, showed a significant difference between the infection and non-infection states, with higher levels when in the presence of infection; Except for TNF, the others showed no difference by *Plasmodium* species.

The level of expression of the six cytokines evaluated in maternal peripheral blood has a behavior similar to that described for placental tissue: five of the six evaluated cytokines show a significant difference between the groups without infection and with infection, except for IL4 (Table 3).

## Correlations between variables

Correlation indicates the strength and direction of a relationship and proportionality between two quantitative variables but does not indicate causal links. The linear correlations between each pair of variables studied (bivariate linear correlations) were evaluated, meaning the linear relationship between them was mediated. If there is no linear relationship, it is possible that there is a non-linear relationship (monotonic curve, non-monotonic curve) or that, in reality, there is no relationship; but only the existence of a linear relationship was evaluated. The

**Table 3. Expression levels of mediators in the placenta by PM group.**

| Variable | Negative | *P. vivax* | *P. falciparum* | p K-W |
|---|---|---|---|---|
| | | Median (Min-Max) | | |
| | n = 38 | n = 20 | n = 25 | |
| Cells in apoptosis | 59 (39–49) | 65(55–86) | 71(59–79) | **≤0,001**[a] |
| | n = 27 | n = 20 | n = 25 | |
| Fas | 5(2.0–4.3) | 5(4.7–6.0) | 5(4.1–5.9) | **≤0,001**[a] |
| FasL | 6(5.2–9.0) | 6(5.3–6.3) | 6(5.1–6.3) | **≤0,001**[a] |
| HIF1α | 1(0.0–2.0) | 1(0.9–1.3) | 1(0.8–2.2) | **≤0,001**[a] |
| Cox1 | 12(0.1–1.5) | 14(1.3–16.0) | 146(10.4–18.7) | **≤0,001**[a] |
| Cox2 | 6(0.1–1.7) | 7(6.2–9.0) | 6(4.8–7.0) | **≤0,001**[a] |
| | n = 9 | n = 13 | n = 9 | |
| C5a | 83 (62.3–113.5) | 97 (57.4–137.9) | 63 (24.0–123.7) | 0.331 |
| | n = 27 | n = 20 | n = 25 | |
| VEGF | 1(0.0–1.4) | 1(0.9–1.2) | 1(0.9–1.2) | **≤0,001**[a] |
| | n = 14 | n = 16 | n = 15 | |
| VEGF-R | 0(0.0–11.7) | 0(0.0–16.3) | 0(0.0–17.9) | 0.675 |
| | n = 6 | n = 6 | n = 13 | |
| IL1 | 1(0.0–1.4) | 2(0.0–8.0) | 2(0.0–72.6) | 0.113 |
| | n = 27 | n = 20 | n = 25 | |
| IL2 | 1(0.4–2.0) | 1(0.9–1.5) | 1(0.8–1.4) | 0.434 |
| IL4 | 1(0.0–2.0) | 2(1.2–2.0) | 1(1.2–2.4) | **≤0,001**[a] |
| | n = 6 | n = 6 | n = 12 | |
| IL6 | 61 (17.0–564.8) | 46 (0.0–253.7) | 50 (0.0–386.1) | 0.584 |
| | n = 5 | n = 6 | n = 11 | |
| IL8 | 1(0.0–3.1) | 1(0.0–2.5) | 1(0.0–12.5) | 0.483 |
| | n = 55 | n = 20 | n = 27 | |
| IL10 | 2(0.0–2.0) | 4(3.0–4.7) | 4(0.1–5.0) | **≤0,001**[a] |
| | n = 45 | n = 20 | n = 26 | |
| IFNγ | 8(0.0–22.0) | 9(8.1–10.0) | 10(0.1–14.0) | **≤0,001**[a] |
| | n = 55 | n = 20 | n = 27 | |
| TNF | 12(0.0–28.1) | 9(11.2–12.8) | 10(3.1–31.2) | **≤0,001**[a] |
| | n = 52 | n = 20 | n = 27 | |
| TGFβ | 2(0.0–2.8) | 2(1.9–3.0) | 2(0.0–3.2) | **≤0,001**[a] |
| | n = 5 | n = 6 | n = 11 | |
| MIP1 | (20.0–10.4) | 1(0.0–5.6) | 2(0.0–19.4) | 0.559 |
| MCP1 | 289 (0.0–417.4) | 316 (38.5–622.3) | 365 (0.0–761.8) | 0.156 |
| | n = 17 | n = 20 | n = 17 | |
| FOXP3 | 1(0.1–0.6) | 1(0.5–2.5) | 1(0.6–1.4) | **≤0,001**[a] |
| | n = 17 | n = 20 | n = 17 | |
| CTLA4 | 2(0.0–4.0) | 3(1.4–9.1) | 3(2.2–5.3) | **≤0,001**[a] |

[a] Dunn´s test with p<0.05 to compare PM negative vs PM by *P. vivax*.

results for the significant bivariate linear correlations (SC) with p<0.100 for the Spearman's Rho coefficient of non-parametric correlation are presented below:

1. **SC between histological events**

    a. **When there is no PM**: the twelve variables/events presented 64 SC; hemorrhages and calcifications are the events with the most SC (10 each), followed by thrombi (9 SC),

capillaries and capillaries per villus (7 each), villi (6 SC), syncytial nodes (5 SC), fibrinoid deposits (4 SC), infarction, edema and abruptio (with 3, 2 and 1 SC, respectively). All 64 are positive SC, except three (gray color), of which two correspond to necrosis with capillaries and thrombi and one to infarction with capillaries per villus.

b. **When there is PM**: the 14 variables (now we must also consider Hz and IE) have 33 SC in PM-V and 24 in PM-F. In the PM-V group, capillaries per villus and hemorrhages have 6 SC each; abruptio, edema, and capillaries have 5 SC each; syncytial nodes have 4 SC. All SC are positive, except the four for infarction with fibrinoid deposits, syncytial nodes, capillaries per villus, and hemorrhages (gray color).

2. **SC between cytokines**

There is a high frequency of SC between the expression levels corresponding to cytokines in placental tissue and maternal peripheral blood, as follows:

a. *Falciparum-PM*: placental tissue cytokines: there is only one SC: IL2 with IL10 rho = -0.427, p = 0.006, n = 40. Maternal peripheral blood cytokines: only one SC: IL10-TNF (+).

b. *Vivax-PM*: placental tissue cytokines: three SC: IL2 with TGFβ rho = -0.475, p = 0.009, n = 29; with INFγ rho = 0.490, p = 0.007, n = 29; with TNF rho = 0.694, p = $\leq$0.001, n = 29.
Maternal peripheral blood cytokines: only one SC: IL10-TGFβ(-).

Among mediators of processes, there are the following SC:

1. *PM-P. falciparum*: there are only four SC, one negative C5a-Fas, and three positives cells in apoptosis with HIF1α and VEGF; Cox1 with VEGF.

2. *PM-P. vivax*: there are no SC. There are many negative correlations that are not significant.

Hemozoin has significant correlations (SC) with these placental cytokines: positive with IL10 and TGFβ and negative with FOXP3. There is no SC between hemozoin and the cytokines evaluated in maternal peripheral blood.

## Discussion

This research on PM in Colombia has two new elements: 1) it examines the largest number of organs; 2) is focused on species of *Plasmodium*. The control group was comprised of apparently healthy women at the time of delivery, according to the medical history data, and with a negative placental blood test with TBS and qPCR for *Plasmodium*. The control group is very high (n = 179) and was determined by a highly sensitive and specific test for *Plasmodium* and, specifically, for *P. vivax*. The women who comprise it belong almost entirely to the socioeconomic groups with the lowest economic capacity; they have been stable residents of the malarious zone for more than 13 years and represent the population of pregnant women exposed to malaria in that region. A variable percentage of women could have been affected by problems such as anemia (hemoglobin <11.0 g/dL), which existed in 25%. It is also very likely that some had intestinal parasites, such as pathogenic soil-transmitted helminths. According to data from our group's ongoing investigation, about 35% of pregnant women residing in the area where this research was carried out have these worms, and 10% have *Plasmodium*-soil-transmitted helminth coinfection. Another problem with a high probability of presence in these pregnant women is chronic malnutrition, which affected 28% of pregnant women in the Urabá Antioqueño and Bajo Cauca Antioqueño between 2005–2010 [20].

In Colombia, by 2005, 5 out of 10 pregnant women suffered from malnutrition, and of these, 40% had a weight deficit, which could have affected the LBW, which reached 6.2% in the country [21]. In Antioquia, by 2010, 53% of households showed food insecurity, 28% of the women had a low gestational body mass index (BMI) [20]), 20% had a low BMI when entering the cohort, and in the third trimester, 48% had inadequate BMI according to pre-pregnancy BMI [22], despite having been in the additional food supply program during pregnancy. These three maternal problems (malnutrition, intestinal parasites, anemia) are interrelated and can also affect the immune system's behavior and be reflected in some of the measurements made in the present work.

The neonate's hemoglobin is lower if the mother has anemia [23, 24], although this anemia does not seem to influence neonatal weight according to studies made in Peru [24, 25]. However, there are contrary reports from the same country about maternal anemia-neonatal weight association. In the third trimester of gestation, the frequency of anemia in mothers of full-term neonates with LBW and adequate birth weight was 52% and 28%, respectively [26]. Anemia in pregnancy is a risk factor for preterm delivery and low birth weight from the third trimester [27].

Some have pointed out that studies that present an association between maternal hemoglobin and neonatal weight have found that neonatal weight varies significantly in relation to this protein's concentration in each trimester of gestation; however, no causal relationship has been established [28].

Low weight gain in pregnant women is associated with intrauterine growth retardation [29, 30]. More specifically, in terms of time, it has been argued that weight gain in the second half of gestation is critical for the fetus's development [31].

It is emphasized that maternal hemoglobin at delivery and neonatal weight present statistically significant differences between the three groups and, above all, between the control and *vivax*-PM groups. It is essential to reiterate that while the neonatal weight in the control group was 3,292 grams, in the *vivax*-PM group, it barely reached 2,841 grams, with both groups having full-term deliveries. It is the first time that, in Colombia, the compromise of neonatal weight by PM and, more specifically, by *vivax*-PM has been demonstrated, beyond doubt; Additionally, this study is focused on *Plasmodium species*, which makes the finding more remarkable. It is essential to emphasize the weight reduction of the neonate and not only on the problem of low birth weight. The weight difference between the children of women with *vivax*-PM compared to the children of women without PM, 451 grams less weight, is sufficient argument to support our claim.

Histological findings and findings of mediators of placental physiological processes represent a good approximation to what happens in this population. Nine of the fourteen placental histological events that were evaluated had significantly different amounts (usually higher amounts) in the groups with PM than the group without PM. It should also be noted that most of these variables differ significantly between the negative-PM and *vivax*-PM groups, but not between the *vivax*-PM and *falciparum*-PM groups. All this points to the pathogenic power of *P. vivax* in the placenta and neonate; similar to that documented by other authors in previous research [32].

The possible pathogenic role of *P. vivax* in PM has been pointed out by various researchers, both in infections detected with TBS and histopathology [9, 10, 33–36] and those indicated as submicroscopic [1–5]. The present work reiterates that statement; despite the most frequent study of *P. falciparum* [37].

As mentioned, inflammation is present in the placentas without infection, but its presence is much higher and significant when there is infection. Something similar happens with the angiogenesis process, regulatory mediators, and most of the examined cytokines. Once again,

the difference between the negative-PM and *vivax*-PM groups stands out, a difference that shows the pathogenic power of this parasite.

The bivariate linear correlations results indicate that immune cell infiltrates are the central points of the correlations found with histological changes in both infected and uninfected placentas. In contrast, the different cells identified by immunohistochemistry do not participate in interactions with histological events, except for CD4+ cells. These correlated events are, essentially, the same in the absence or presence of infection. It is possible that the reduced list of mediators evaluated among the many who may participate during inflammation, hypoxia, and apoptosis, among others, could be an explanation for the absence of differences between correlations in the absence or presence of placental plasmodial infection. The correlations observe here allow us to have foundations to advance the understanding of the histological changes and physiological processes that occur during placental malaria. The minimum requirements to measure a correlation are of several classes, and it also must be borne in mind that correlation does not imply causation [38, 39].

The examination of placentas with *vivax*-PM allows an important approach to studying this entity in our country and the international context. Much of the analysis was focused on comparing *vivax*-PM with the absence of PM because there is insufficient argumentation about the pathogenic capacity of this species in the case of the placenta.

A special issue that must be highlighted from this work is the comprehensive approach to the study of placental malaria, in the sense of covering different problems: basic epidemiology, histopathology, immunohistochemistry, cytokines, and the mediators of pathophysiological processes involved in it.

This situation prompts us to emphasize the necessity of using, in"The Region", an intermittent preventive treatment in pregnancy (IPTp) with drugs such as sulfadoxine-pyrimethamine (SP) or amodiaquine plus sulfadoxine-pyrimethamine (SP), as is done in Africa with SP, with excellent results because it reduces the magnitude of anemia in mothers and children and increases neonatal weight [40, 41]. In areas of stable malaria transmission, since 2004, WHO has supported a three-pronged MIP approach: (1) intermittent preventive treatment with sulfadoxine-pyrimethamine; (2) use of insecticide-treated bed nets; and (3) effective case management [42]. A recent meta-analysis analyzed the microeconomic evaluations of pregnancy associated malaria (PAM) reported in the scientific literature. All the interventions were highly cost-effective, which demonstrates the importance of including prevention, care, and control resources for PAM as a priority in health sector budgets. The IPTp with SP was the most frequent strategy [43].

## Conclusion

The study data show that infection of the placenta, both with *P. falciparum* and with *P. vivax*, damages this organ and causes significant alteration of various physiological processes, which cause maternal anemia and a reduction in neonatal weight in degrees that are statistically and clinically significant. For this reason, it is necessary that the search for plasmodial infection in pregnant women and placenta goes from passive to active screening and uses tools with adequate diagnostic capacity (sensitivity, specificity).

## Supporting information

**S1 Text. Definition of histological events in the placenta.**
(DOCX)

## Author Contributions

**Conceptualization:** Jaime Carmona-Fonseca, Jaiberth Antonio Cardona-Arias.

**Data curation:** Jaime Carmona-Fonseca.

**Formal analysis:** Jaime Carmona-Fonseca, Jaiberth Antonio Cardona-Arias.

**Funding acquisition:** Jaime Carmona-Fonseca.

**Investigation:** Jaime Carmona-Fonseca, Jaiberth Antonio Cardona-Arias.

**Methodology:** Jaime Carmona-Fonseca, Jaiberth Antonio Cardona-Arias.

**Project administration:** Jaime Carmona-Fonseca.

**Resources:** Jaime Carmona-Fonseca.

**Software:** Jaime Carmona-Fonseca, Jaiberth Antonio Cardona-Arias.

**Supervision:** Jaime Carmona-Fonseca.

**Validation:** Jaime Carmona-Fonseca, Jaiberth Antonio Cardona-Arias.

**Visualization:** Jaime Carmona-Fonseca, Jaiberth Antonio Cardona-Arias.

**Writing – original draft:** Jaime Carmona-Fonseca, Jaiberth Antonio Cardona-Arias.

**Writing – review & editing:** Jaime Carmona-Fonseca, Jaiberth Antonio Cardona-Arias.

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
