## [Decision Letter · Decision Letter 0]

4 Nov 2021

PONE-D-21-30845Placental malaria caused by Plasmodium vivax or P. falciparum in Colombia: histopathology and mediators in placental processesPLOS ONE

Dear Dr. Cardona-Arias,

Thank you for submitting your manuscript to PLOS ONE. After careful consideration, we feel that it has merit but does not fully meet PLOS ONE’s publication criteria as it currently stands. Therefore, we invite you to submit a revised version of the manuscript that addresses the points raised during the review process.

The manuscript deals with placental malaria with different parameters. The authors said they collected data by a questionnaire and the study includes 15 years of study, so, if they applied this questionnaire during this period? Beside the manuscript needs language editing in some parts. There are some issues addressed by the reviewers.  

We look forward to receiving your revised manuscript.

Kind regards,

Shawky M. Aboelhadid, PhD

Academic Editor

PLOS ONE

Journal Requirements:

Reviewers' comments:

Reviewer's Responses to Questions

**Comments to the Author**

1. Is the manuscript technically sound, and do the data support the conclusions?

Reviewer #1: Yes

2. Has the statistical analysis been performed appropriately and rigorously? 

Reviewer #1: Yes

3. Have the authors made all data underlying the findings in their manuscript fully available?

Reviewer #1: Yes

4. Is the manuscript presented in an intelligible fashion and written in standard English?

Reviewer #1: Yes

5. Review Comments to the Author

Reviewer #1: There are a number of issues that the authors need to address to improve the clarity of their manuscript:

1. Abstract: needs attention to English grammatical errors, and also some adjustment to incorporate findings relating to infection with P. falciparum. As it stands, the abstract focuses solely on findings related to infection with P. vivax.

2. How accurate are the estimates of term based on date of last menstruation? In the same context, the Methods refer to term falling between 38-42 weeks, whilst the legend to Figure 1 states 37-42 weeks. There needs to be consistency here. Also in the same context, the authors refer to malaria episodes during pregnancy. How were these documented/recorded? Were such events diagnosed and treated accordingly, and if so, did the authors account for possible bias in the data collected from women with these specific histories?

3. It is unclear (i) how different blood samples (perupheral and placental) were collected - venipuncture for peripheral blood is standard, but more detail of how placental blood was collected needs to be given; (ii) there is no description of the methods used for collection and storage of placental tissue samples - the relevant Methods section refers to the PCR-based methods used with blood samples; (iii) in which samples cytokine expression levels were measured - the Methods refer to analyses in placental tissue samples, whilst the Results section refers to SC correlations using data derived from analyses with cytoline expression levels in maternal peripheral blood samples

4. In the Methods, the authors state that, for immunohistochemical, cytokine and physiological process evaluations, 25 samples per group were selected at random. That being said, it would be informative to know if the general characteristics of these individuals were similar to those of the whole group. In the same context, Table 3 presents data for the cytokine expression levels for which the group sizes vary markedly both above and below the number of 25 per group. This should be clarified, and the authors should also explain why the immunohistochemical results for immunomarker expression (CD4/CD8 etc) are presented in Table 2 with histopathological data.

5. It is unusual to include citations to non-English bibliography, many, it appears, published in journals that are not international in scope, and actually unacceptable to therein include citations that cannot be accessed by reviewers or readers.

6. The authors have chosen to ignore several relevant articles in the literature. To cite just 2 obvious examples: (i) Chaikitgosiyakul et al Malaria Journal 2014; (ii) Muehlenbachs et al Journal of Infectious Diseases 2010

6. PLOS authors have the option to publish the peer review history of their article (what does this mean?). If published, this will include your full peer review and any attached files.

Reviewer #1: **Yes: **Adrian JF Luty

---

## [Author Response · Author response to Decision Letter 0]

17 Dec 2021

Medellín, Colombia, December 17th, 2021

PhD Shawky M. Aboelhadid 

Academic Editor

PLOS ONE

Reference: Response to reviewers PONE-D-21-30845 Placental malaria caused by Plasmodium vivax or P. falciparum in Colombia: histopathology and mediators in placental processes

Kind regards,

We thank you for your valuable evaluation, which allows us to improve the quality of the manuscript. Through this letter we inform the realization of all the changes suggested by the journal and reviewers. Below we describe in detail the changes made consistent with each observation of reviewers

Editor' comments

Observation 1. The manuscript deals with placental malaria with different parameters. The authors said they collected data by a questionnaire and the study includes 15 years of study, so, if they applied this questionnaire during this period?

R/ Yes. It is a questionnaire to extract variables from the medical chart of each participant, which did not change during the study period (in Colombia, the clinical history of gynecological and obstetric data has not changed in recent decades).

Observation 2. Beside the manuscript needs language editing in some parts. There are some issues addressed by the reviewers. 

R/ We make editing corrections throughout the entire text

Journal Requirements

Observation 1. Please ensure that your manuscript meets PLOS ONE's style requirements, including those for file naming. The PLOS ONE style templates can be found at https://journals.plos.org/plosone/s/file?id=wjVg/PLOSOne_formatting_sample_main_body.pdf and https://journals.plos.org/plosone/s/file?id=ba62/PLOSOne_formatting_sample_title_authors_affiliations.pdf

R/ The adjustments were made.

Observation 2. We note that you have stated that you will provide repository information for your data at acceptance. Should your manuscript be accepted for publication, we will hold it until you provide the relevant accession numbers or DOIs necessary to access your data. If you wish to make changes to your Data Availability statement, please describe these changes in your cover letter and we will update your Data Availability statement to reflect the information you provide.

R/ We agree with the journal's message about “we will hold it until you provide the relevant accession numbers or DOIs necessary to access your data”.

Observation 3. Please include captions for your Supporting Information files at the end of your manuscript, and update any in-text citations to match accordingly. Please see our Supporting Information guidelines for more information: http://journals.plos.org/plosone/s/supporting-information.

R/ The change was made

Reviewers' comments

Observation 1. Abstract: needs attention to English grammatical errors, and also some adjustment to incorporate findings relating to infection with P. falciparum. As it stands, the abstract focuses solely on findings related to infection with P. vivax.

R/ The change was made

Observation 2. How accurate are the estimates of term based on date of last menstruation? In the same context, the Methods refer to term falling between 38-42 weeks, whilst the legend to Figure 1 states 37-42 weeks. There needs to be consistency here. Also in the same context, the authors refer to malaria episodes during pregnancy. How were these documented/recorded? Were such events diagnosed and treated accordingly, and if so, did the authors account for possible bias in the data collected from women with these specific histories?

R/ The estimates of term based on date of last menstruation are precise, this coincides with the gestational age according to the ultrasound report (but in this study the ultrasound data was not obtained for all the subjects).

We make the correction of the number 38.

The information about previous cases of malaria during the current pregnancy was by self-report (according to the self-report of the pregnant women, all previous cases were treated, and therefore no bias is incurred).

Observation 3. It is unclear (i) how different blood samples (perupheral and placental) were collected - venipuncture for peripheral blood is standard, but more detail of how placental blood was collected needs to be given; (ii) there is no description of the methods used for collection and storage of placental tissue samples - the relevant Methods section refers to the PCR-based methods used with blood samples; (iii) in which samples cytokine expression levels were measured - the Methods refer to analyses in placental tissue samples, whilst the Results section refers to SC correlations using data derived from analyses with cytoline expression levels in maternal peripheral blood samples.

R/ i) Clarification regarding the taking of placental blood samples (lines 137 - 140) was made.

ii) The storage process is not described, since all the placental samples were processed immediately they were collected, and what was collected for this study were the results of these analyzes in a databse. However, it is important to clarify that the research group conserves the blood samples and histopathology, with the quality standards that are required for correct fixation, conservation and preservation (as indicated in other parts of the methods, for example lines 214-216).

iii) To the cytokines indicated in the lines 210-213. Most of the results refer to the placenta as a unit of analysis and not as a biological sample (in the latter case, the samples correspond to those described in the methods for blood or placental tissue)

Observation 4. In the Methods, the authors state that, for immunohistochemical, cytokine and physiological process evaluations, 25 samples per group were selected at random. That being said, it would be informative to know if the general characteristics of these individuals were similar to those of the whole group. In the same context, Table 3 presents data for the cytokine expression levels for which the group sizes vary markedly both above and below the number of 25 per group. This should be clarified, and the authors should also explain why the immunohistochemical results for immunomarker expression (CD4/CD8 etc) are presented in Table 2 with histopathological data.

R/ The basic characteristics of the subgroup are similar to the total studied (this was not added to the text due to this unnecessarily extend the manuscript).

Regarding the variability in the sample size of the subgroups, this is explained in two matters: some samples did not meet the pre-analytical conditions, which led to a reduction in the number analyzed, and in other cases it was possible to obtain financial resources to increase the sample size. In general, these subgroup analyzes are exploratory and therefore we were unable to apply robust parameters for a sample size calculation, given that there are no analytical studies that would allow us to establish a minimum expected difference between the three groups compared.

Lymphocytes were included in table two, since conceptually, cells are part of the histological component.

Observation 5. It is unusual to include citations to non-English bibliography, many, it appears, published in journals that are not international in scope, and actually unacceptable to therein include citations that cannot be accessed by reviewers or readers.

R/ This is due to two facts: i) low number of similar studies, ii) need to make comparisons with similar populations that have been described in publications with languages other than English.

Observation 6. The authors have chosen to ignore several relevant articles in the literature. To cite just 2 obvious examples: (i) Chaikitgosiyakul et al Malaria Journal 2014; (ii) Muehlenbachs et al Journal of Infectious Diseases 2010

R/ These were added to the new version of the manuscript.

Kind regards,

The authors.

---

## [Decision Letter · Decision Letter 1]

12 Jan 2022

Placental malaria caused by Plasmodium vivax or P. falciparum in Colombia: histopathology and mediators in placental processes

PONE-D-21-30845R1

Dear Dr. Cardona-Arias,

We’re pleased to inform you that your manuscript has been judged scientifically suitable for publication and will be formally accepted for publication once it meets all outstanding technical requirements.

Kind regards,

Shawky M Aboelhadid, PhD

Academic Editor

PLOS ONE

Additional Editor Comments (optional):

Reviewers' comments:

Reviewer's Responses to Questions

**Comments to the Author**

1. If the authors have adequately addressed your comments raised in a previous round of review and you feel that this manuscript is now acceptable for publication, you may indicate that here to bypass the “Comments to the Author” section, enter your conflict of interest statement in the “Confidential to Editor” section, and submit your "Accept" recommendation.

Reviewer #1: All comments have been addressed

2. Is the manuscript technically sound, and do the data support the conclusions?

Reviewer #1: (No Response)

3. Has the statistical analysis been performed appropriately and rigorously? 

Reviewer #1: (No Response)

4. Have the authors made all data underlying the findings in their manuscript fully available?

Reviewer #1: (No Response)

5. Is the manuscript presented in an intelligible fashion and written in standard English?

Reviewer #1: (No Response)

6. Review Comments to the Author

Reviewer #1: (No Response)

7. PLOS authors have the option to publish the peer review history of their article (what does this mean?). If published, this will include your full peer review and any attached files.

Reviewer #1: **Yes: **Adrian J F Luty

---

## [Editor Report · Acceptance letter]

14 Jan 2022

PONE-D-21-30845R1 

Placental malaria caused by *Plasmodium vivax* or *P. falciparum* in Colombia: histopathology and mediators in placental processes 

Dear Dr. Cardona-Arias:

I'm pleased to inform you that your manuscript has been deemed suitable for publication in PLOS ONE. Congratulations! Your manuscript is now with our production department. 

Kind regards, 

on behalf of

Professor Shawky M Aboelhadid 

Academic Editor

PLOS ONE